# Pareto optimality between growth-rate and lag-time couples metabolic noise to phenotypic heterogeneity in *Escherichia coli*

Diego Antonio Fernandez Fuentes [1], Pablo Manfredi[2], Urs Jenal [2] & Mattia Zampieri [1✉]

Despite mounting evidence that in clonal bacterial populations, phenotypic variability originates from stochasticity in gene expression, little is known about noise-shaping evolutionary forces and how expression noise translates to phenotypic differences. Here we developed a high-throughput assay that uses a redox-sensitive dye to couple growth of thousands of bacterial colonies to their respiratory activity and show that in *Escherichia coli*, noisy regulation of lower glycolysis and citric acid cycle is responsible for large variations in respiratory metabolism. We found that these variations are Pareto optimal to maximization of growth rate and minimization of lag time, two objectives competing between fermentative and respiratory metabolism. Metabolome-based analysis revealed the role of respiratory metabolism in preventing the accumulation of toxic intermediates of branched chain amino acid biosynthesis, thereby supporting early onset of cell growth after carbon starvation. We propose that optimal metabolic tradeoffs play a key role in shaping and preserving phenotypic heterogeneity and adaptation to fluctuating environments.

[1] Institute of Molecular Systems Biology, ETH Zürich, Zürich, Switzerland. [2] Biozentrum, University of Basel, Basel, Switzerland. ✉email: zampieri@imsb.biol.ethz.ch

Heterogeneity between individual bacterial cells growing under macroscopically identical conditions can have important functional consequences[1]. While not always beneficial[2,3], population diversification is a key mechanism to adapt to fluctuating environments[4], in that it can allow genotypes to persist in the face of adverse conditions. For example, growth rate heterogeneity in bacterial populations contributes to survival upon exposure to antibiotics[5,6], and differences in the onset of cell division can favor tolerance and evolution of antibiotic resistance[7–9]. A number of different molecular mechanisms can give rise to phenotypic heterogeneity[10–12]. Typically, these are cellular processes such as stochastic gene expression[13,14] and stochastic partitioning of molecules at cell division[15], which ultimately manifest in the variation of protein copy numbers per cell. Recent experimental evidence showed that fluctuations in the expression of growth-limiting enzymes can be responsible for growth fluctuations and the transmission of noise to other genes[10]. While mounting single-cell studies have shown gene expression to vary within a population of genetically identical cells under constant conditions[13,16,17], which cellular processes are predominantly affected by noisy gene expression and whether noise propagation is shaped by optimality-based principles are still open questions.

Despite the expectations that stochasticity of biochemical reactions should have negligible effect, theoretical models demonstrate how noise in transcription and translation of metabolic enzymes can affect levels of metabolic reactants[18,19]. However, whether and how fluctuations in metabolite levels can permeate the metabolic state of a cell and affect the ability to adapt to changing environments is unclear. While technological advances enabled monitoring levels of metabolites at single cell[20,21], directly probing cell-to-cell variability in the turnover of metabolites remains challenging[22], ultimately hampering the study of metabolic heterogeneity and its functional implications. By developing an assay to directly measure variability in the respiratory rates among colonies of *Escherichia coli* growing on solid media, we show that noisy transcriptional regulation of lower glycolytic and citric acid (TCA) cycle enzymes can be readily transmitted to metabolic heterogeneity. Surprisingly, we found that individual colonies within an isogenic population exhibit largely different respiratory activities, and that while costly[23], increased respiratory activity facilitates the early onset of cell growth after starvation by preventing the accumulation of toxic intermediates. We propose that variability in respiro-/fermentative metabolism can be fitness invariant, allowing cells to maintain a greater variation in enzyme expression and potentially employ diverse adaptive strategies to environmental changes.

## Results

### Enzymes involved in oxidative-reductive reactions exhibit large cell-to-cell variability compared to proteins with a similar abundance.

To find cellular processes that exhibit large cell-to-cell variability, we analyzed previously published single-cell proteome data of *E. coli*[13] (Fig. 1A). In prokaryotes, as in unicellular eukaryotes, variability in gene expression and protein levels among cells (i.e., noise) is inversely proportional to the mean expression level of the population[13,24,25]. However, because on average, in *E. coli* essential genes encode for highly expressed proteins ($p$-value = 2.64e−10) (Fig. 1B, C)[13], we hypothesized that noise in abundant proteins, even if modest, may have important consequences for phenotypic heterogeneity. To systematically compare noise among diversely expressed proteins, we estimated the deviating noise, here defined as the deviation of each protein from the average noise levels of proteins with similar expression levels (Figs. 1A and S1, Supplementary Data 1). Next,

we performed a gene set enrichment analysis and compared biological processes that are enriched for proteins with a large standard (i.e., squared coefficient of variation) or deviating noise (Fig. 1D, Supplementary Data 1). While the most significant ($q$ value ≤ 1e−4) processes affected by standard noise levels consist of lowly expressed genes involved in DNA repair, DNA recombination, and chemotaxis (Fig. 1B–D), we found that proteins with the largest deviating noise levels are enriched for metabolic genes involved in oxidative-reductive reactions (Fig. 1D). More specifically, we found that genes encoding for enzymes in central metabolism (e.g., *aceE*) exhibit larger than expected cell-to-cell variability relative to other proteins with similar expression levels (Figs. 1A and S1). On the other hand, essential genes on average exhibited low deviating noise levels ($p$-value ≤ 0.05), indicating that there may be selective pressures for noise reduction[26]. These results suggest that, despite the key role in bacterial fitness of proteins involved in central metabolism, fluctuations in their expression, translation, or degradation may have been preserved by evolution and can be a major source of phenotypic heterogeneity. To test if deviations of proteins from mean-noise levels could be the results of stochasticity in their gene expression regulation, we searched for transcription factors (TFs) regulating the expression of proteins with large deviating noise. We found that Cra, a key regulator of flux in lower glycolysis[27], is significantly enriched ($p$ value ≤ 1e−4 Bonferroni-corrected threshold) for targets that exhibit high deviating noise levels (Fig. 1E). Altogether, these analyses suggest that noise in protein abundance can potentially translate into large metabolic cell-to-cell variability and that such heterogeneity can be at least partially explained by a few TFs.

### Noise propagation to respiro-/fermentative metabolism.

Our results are consistent with other studies reporting on large variations in enzyme levels[10,28]. However, whether and which noisy enzymes are able to cause fluctuations in the rates of the corresponding enzymatic reactions, and hence may affect phenotypic heterogeneity, is still unclear. To address this question, we integrated bulk measurements of protein copy-numbers[29] and metabolic fluxes[30] in *E. coli* growing in minimal media with seven different carbon sources. By quantifying the linear dependency between changes in the copy number of 40 enzymes and 25 rates of the corresponding metabolic reactions, we found that changes in the abundance of enzymes in lower glycolysis (e.g., Pdh, GapA, Pgk) and citric acid (TCA) cycle (e.g., GltA) directly scale with changes in fluxes (Figs. 1F and S2, Supplementary Data 1). Hence, expression noise in these enzymes can directly translate into cell-to-cell flux variability and overall is likely to cause large changes in respiro-/fermentative metabolism.

Experimentally validating noise propagation from enzyme expression to metabolism remains challenging[31,32]. Currently, experimental tools probing the metabolism of single cells are limited to the monitoring of metabolite abundance[33,34], and so far direct measurements of metabolic fluxes (i.e., metabolite's turnover rate) at the single-cell level are lagging behind. As a result, much less is known about heterogeneity at the level of metabolism, and whether such variability can have direct functional consequences. To overcome this problem, instead of single-cell analytical methods we developed a multiparametric assay that monitors phenotypic and metabolic heterogeneity of thousands of colonies-forming *E. coli* growing on solid agar media (Fig. S3). Our approach was inspired by studies using a scanner array coupled with image analysis to monitor colony-growth[35,36]. For each colony, we estimated the maximum growth rate and lag time by fitting colony area over time with a Gompertz growth function[37] (Fig. S3). In addition, we added 2,3,5-

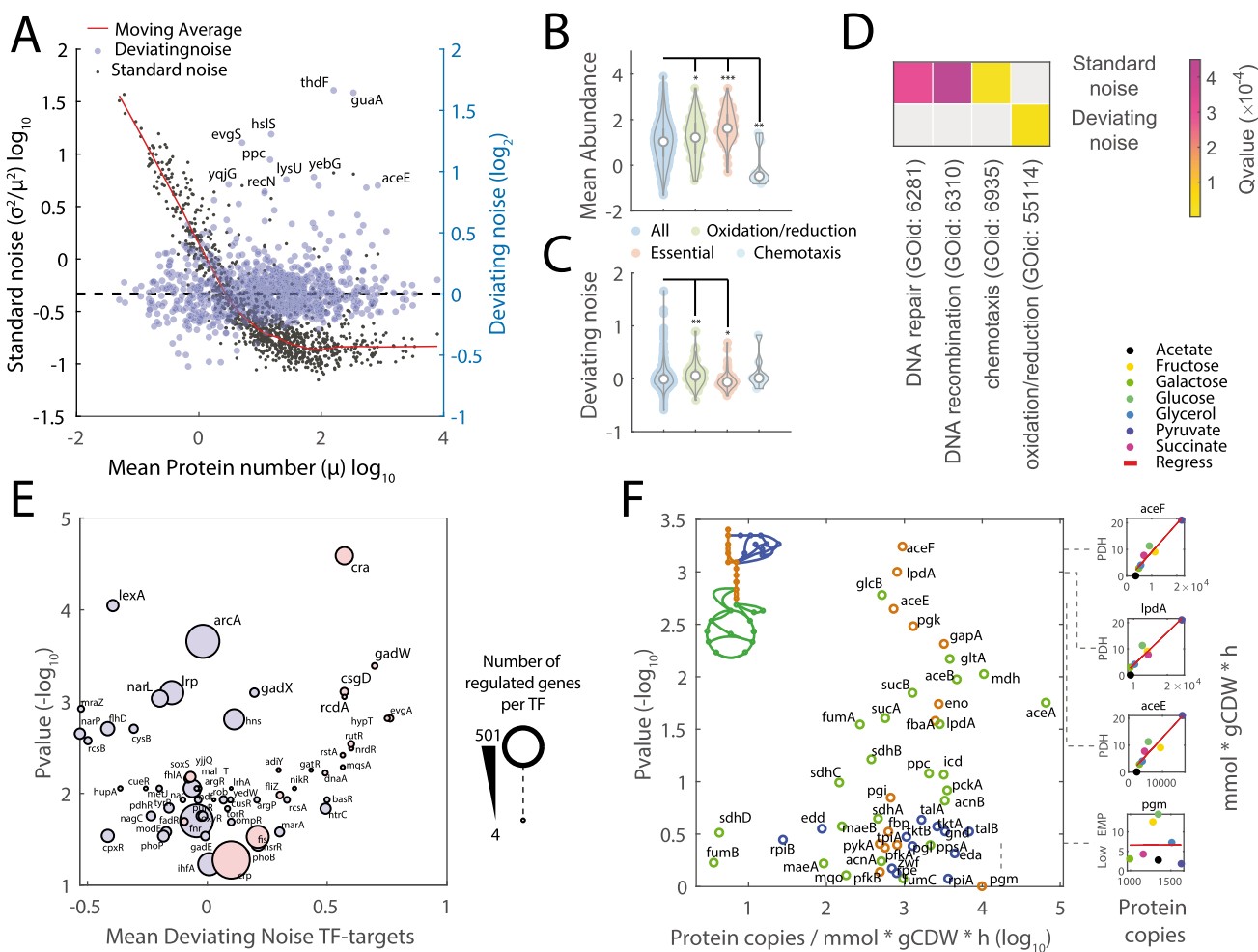

**Fig. 1 Noise propagation to metabolism. A** Time independent protein ($p$) abundance and noise from single-cells of *E. coli*[13]. Average copy number ($\mu_p$) vs. protein expression noise ($\eta = \sigma_p^2/\mu_p^2$, where $\sigma_p$ is the standard deviation) of 1018 proteins (gray dots). A moving average smoothing method is used to estimate average noise levels at different protein abundance ($\bar{\eta}_p$) (red line). Deviating noise (blue dots) is calculated as the ratio between the noise of individual proteins and average noise levels of proteins with similar abundance ($\varepsilon_p = \eta_p/\bar{\eta}_p$) (Supplementary materials and Supplementary Data 1). Highlighted with gene names, proteins that exhibit the highest deviating noise levels. The black dashed line corresponds to zero deviating noise. **B**, **C** Distributions of average protein abundance and deviating noise levels. Asterixis indicate significance, two-tailed *t*-test: *$p$ value ≤ 0.05, **$p$ value ≤ 0.01, ***$p$ value ≤ 0.001. **D** Gene ontology enrichment (http://geneontology.org/) reporting biological processes that are significantly (permutation test with Storey correction $q$ value < 0.001) enriched for proteins with high deviating or standard noise levels. **E** For each transcription factor (TF) we estimated the average protein deviating noise of target regulated genes and the significance of the enrichment for target regulated genes with large deviating noise levels. TFs regulating proteins with a positive or negative median deviating noise is colored in red and blue, respectively. Size of the dot scales with the number of target-regulated genes. **F** For each enzyme in central metabolism we used ordinary least square regression analysis to estimate the proportionality and significance between protein copy number[29] and corresponding flux rates[30] across seven different conditions. Enzymes are colored by pathway: orange-glycolysis, blue-pentose phosphate pathway, green-TCA, as indicated by the small schematic on the upper left corner (see Fig. S1 for full details).

triphenyl-2H-tetrazolium chloride (TTC) to the agar medium. Commonly used in viability assays, TCC is a redox-sensitive dye that is reduced by electron transfer from the respiratory chain with the formation of 1,3,5-triphenylformazan (TPF), a water-insoluble red fluorescent intracellular formazan[36,38] (Fig. S3). By measuring the rate at which each colony turns red, TCC allowed us to directly measure the overall cellular reduction rate, and hence to estimate the respiratory rate in single colonies (Fig. S3, Supplementary Data 1). Overall, we found large variability in growth kinetic parameters among colonies growing on Luria Bertani (LB) and glucose M9 agar plates (Fig. 2A–D) (coefficient of variation (CV) ~40 and 15% for growth rate and lag time, respectively measured in three biological replicates). Moreover, consistent with proteome-based predictions, we found considerable variability also in respiratory rates (Fig. 2E) (CV ~18 and

30% in M9 and LB, respectively). Notably, while on average TCC inhibits bacterial growth, it does not affect colony-to-colony variability (Fig. S3). Our analysis of protein noise levels in *E. coli* (Fig. 1) predicted that variability in respiratory activity relates to noise in the regulation of lower glycolytic enzymes. We hypothesized that using an inducible promoter to control Cra overexpression would reduce expression noise in its target genes and overall result in lower colony-to-colony variations. To verify our hypothesis, we compared the heterogeneity in growth and respiratory rates among colonies of wild-type, Δ*cra*, and *cra* overexpression mutants (*cra*+) (Supplementary Data 1). Consistent with our expectations, we observed that Cra over-expression induces a significant ($p$ value ≤ 0.01) reduction in the variability of respiratory rates as well as lag times (Figs. 2F–H and S4), indicating that even small variations of the levels of lower

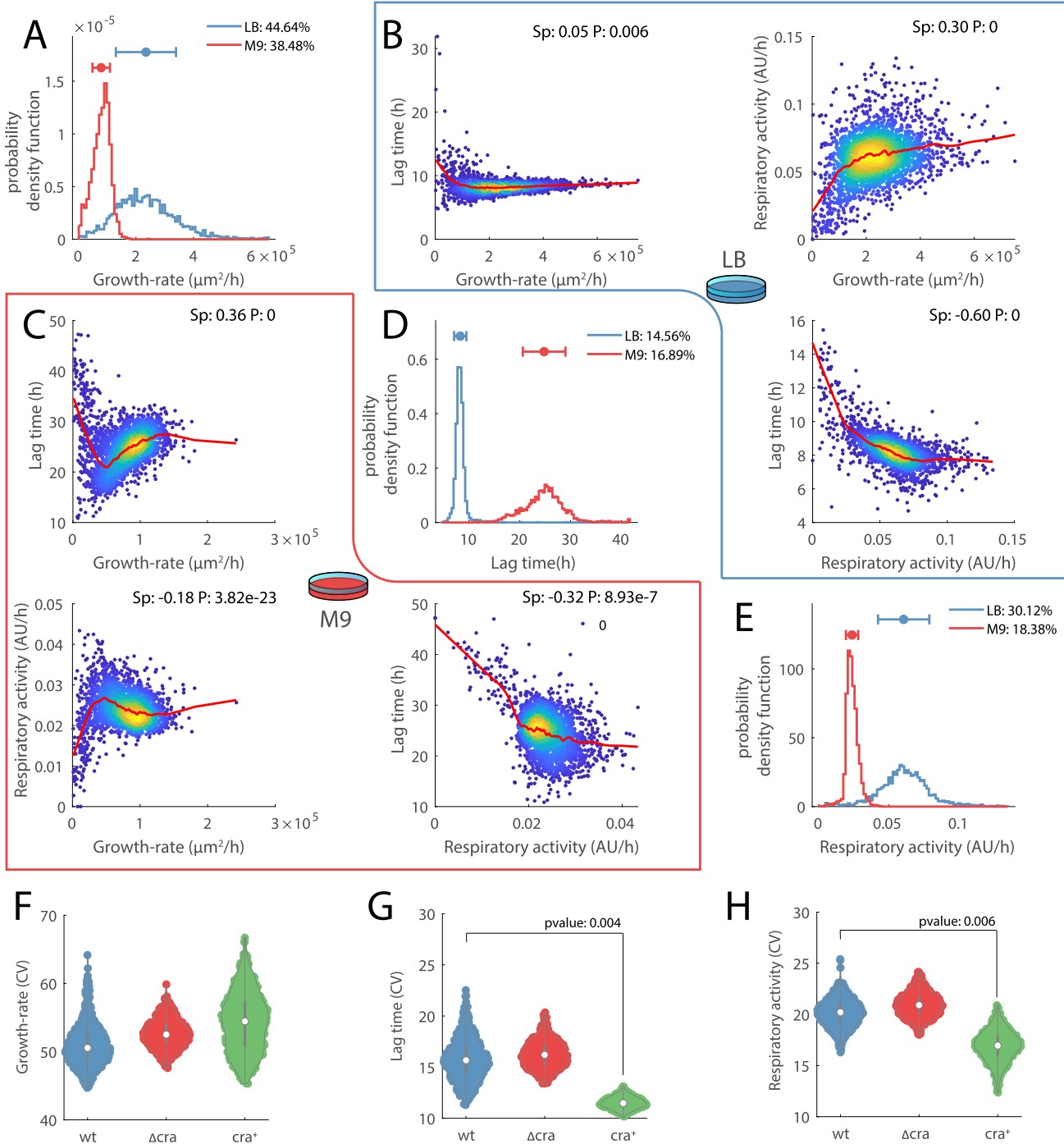

**Fig. 2 Colony-to-colony variation. A, D, E** Distribution of maximum growth rates, lag times, and respiratory rates, estimated for 5485 colonies of wild-type E. coli growing in LB (red) and glucose M9 (blue) agar plates (Supplementary Data 1). Estimates were derived from colonies grown independently in three different plates with LB and three plates with M9 glucose solid media. **B, C** 2D density plots in which each dot represents one colony and the red line a locally weighted smoothed average (i.e., lowess function). On the upper right corner of each panel, we reported the Spearman correlation (Sp) and corresponding p value (P, testing the hypothesis of no correlation against the alternative hypothesis of a nonzero correlation using the Spearman's Rho test) between growth rate, lag time and respiratory activity in LB (**B**) and glucose M9 (**C**). **F, G, H** Distribution of CVs for growth parameters estimated from 1000 random selections of 400 colonies from wild-type (blue), Δcra (red), and cra+ mutant (green) (Fig. S4). Reported significance was estimated from a two-tailed t-test.

glycolytic enzymes can translate in large phenotypic differences. Somehow unexpectedly, noise reduction was not observed at the growth rate level (Fig. 2F).

**Pareto optimal tradeoffs between respiration and fermentation.** If noise in gene expression of enzymes in central

metabolism is responsible for fluctuations in respiratory rates, the next question is whether and how such metabolic fluctuations propagate and relate to phenotypic heterogeneity. To address this question, colony-to-colony variability in growth rate, lag time and respiratory activity of E. coli growing on LB or glucose M9 agar plates are related to each other by estimating their pair-wise

Spearman correlation (Fig. 2B, C). Overall, we found that respiratory activity and growth rates are poorly correlated, and no correlation was observed between lag time and maximum growth rate in the LB medium, consistent with previous studies[39] (Fig. 2B, C). Surprisingly, we found a strong correlation between respiratory activity and lag time (Fig. 2B, C), suggesting that the higher the respiratory activity the shorter is the time needed for cells to start duplicating. To test whether this phenomenon generalizes to largely diverse conditions and growth rates, we monitored growth and metabolic activity in thousands of colonies (i.e., ~8000) in the presence of 13 different perturbing agents on independent LB plates. These are mostly antibiotics interfering with different cellular processes, such as protein/RNA (i.e., Chloramphenicol, Rifampicin, Erythromycin, Tetracycline) and cell wall synthesis (i.e., Bacitracin, Fosfomycin, Ampicillin, Cefaclor), DNA replication (i.e., Ciprofloxacin, Nalidixic acid, Nitrofurantoin), and ATP biosynthesis (i.e., Carbonyl cyanide, Sodium Azide). To test different concentrations of the same antibiotic in one plate, we spotted the center of the dish with the antibiotic (Supplementary Data 1), and let it diffuse for 3 h before inoculation. This way, the closer the colony is to the center of the plate, the higher is the concentration of the antibiotic (Fig. 3A). This approach allowed us to test a continuous range of growth rates, lag times, and respiratory activities in a single petri dish (Fig. 3A–C). Remarkably, the strong correlation between respiratory rate and lag time held true in all conditions tested (Fig. S5), suggesting that the link between heterogeneity in respiratory activity and the onset time of colony expansion might

reflect a fundamental mechanism at the basis of the decision-making process of cell division.

Mounting evidence has shown that respiration, although being a more efficient way for *E. coli* to utilize available nutrients for energy generation, requires larger investments in proteins[23]. Hence, for the same ATP production, fermentation consumes more carbon but requires smaller investment in catalytic machinery allowing faster growth[23,40]. Coherently, the fastest growing colonies in M9 with glucose exhibited relatively low respiratory activities (Fig. 2C). On the other hand, colonies with the highest respiratory activities, while generally showing lower growth rates, featured the shortest lag times (Fig. 2B, C). This finding suggests that maximization of growth rate and minimization of lag time are competing cellular objectives and that the regulation of respiro-/fermentative metabolism can favor one objective over the other—i.e., while fermentation allows for faster growth[23], respiration fosters quicker initiation of cell growth. Moreover, instead of a large population of equally fast-growing individuals and a small unfit sub-population that could ensure survival upon unfavorable environmental changes (e.g., persisters), we observed a continuum of colonies exploiting different metabolic strategies. How can this metabolic diversity be maintained? While in a given constant environment, only one strategy is optimal, in a naturally fluctuating environment the advantages of short lag time or fast growth rate might average out. We hypothesized that variability in respiro-/fermentative metabolism is fitness-neutral—i.e., cells can be equally optimal in spite of different metabolic strategies. Colonies that find an

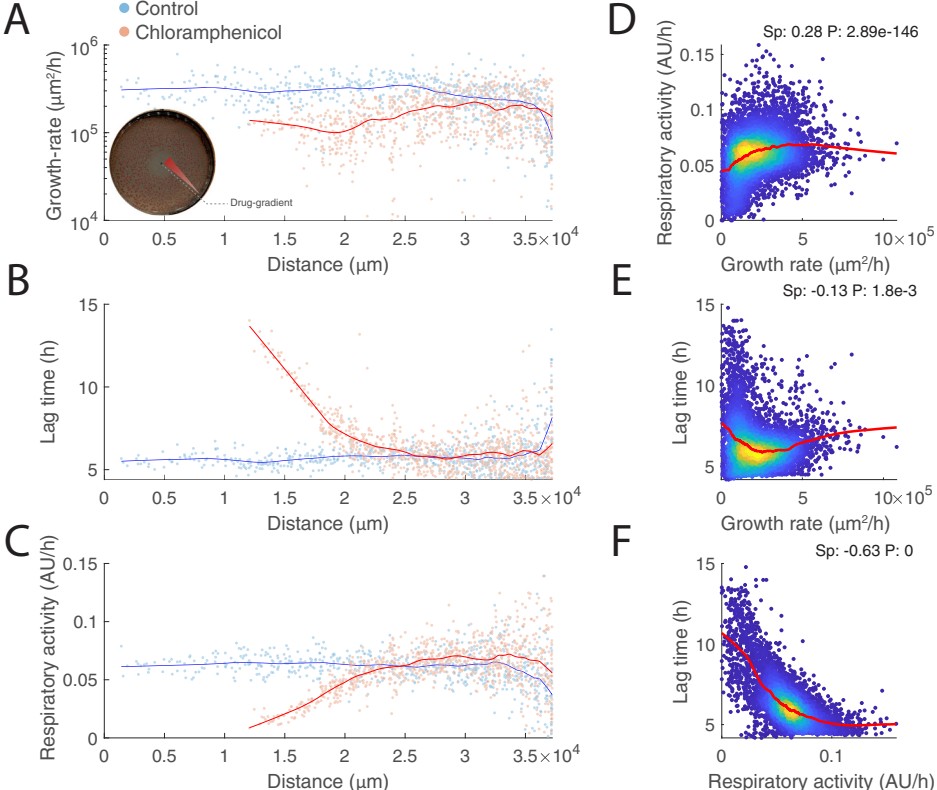

**Fig. 3 Respiratory activity vs. lag time. A–C** Growth rates, lag times, and respiratory activities of individual colonies are plotted against their distance from the center of the plate (*x*-axis). Each dot represents a colony, growing in the same plate with only LB solid medium (blue), or with the addition of chloramphenicol spotted in the center of the plate. Blue (control) and red (Chloramphenicol treatment) thick lines are smoothed values obtained from a locally weighted smoothing function (i.e., lowess function). **D–F** 2D density plots of estimates from all 8718 individual colonies' growth rate, lag time, and respiratory activity across all 14 tested conditions (i.e., one plate per condition) (Supplementary Data 1). Red thick lines are smoothed values obtained from a locally weighted smoothing function. On the upper right corner of each panel, we reported the Spearman correlation (Sp) and corresponding p value (*P*), testing the hypothesis of no correlation against the alternative hypothesis of a nonzero correlation using the Spearman's Rho test.

optimal compromise between two competing objectives are called Pareto optimal[41]—i.e., a colony is Pareto optimal if there exists no other colony that is at least as good in all objectives, but strictly better in at least one objective. By using a simple exponential model of bacterial growth, one can derive that in equally fit cells —i.e., cells able to reach the same number of divisions in a fixed period of time—growth rate and lag time are inversely related (Supplementary text). Hence, according to our hypothesis, the space of feasible growth rates and lag times should be delimited by a Pareto front[41] approximating an inverse relationship between the two. Moreover, natural selection of optimal fitness will force colonies to operate metabolism in the proximity of the optimal tradeoffs[42], and hence colonies shall not randomly occupy the space of feasible growth rates and lag times. Moreover, because of the already known direct proportionality between fermentative metabolism and growth rate[23], we expect that in Pareto optimal colonies, growth rate negatively scales with the respiratory rate (Supplementary text: lag time vs. growth rate). Our theory is consistent with growth kinetics observed in wild-type *E. coli* colonies growing on glucose as the sole carbon source (Fig. 4A). We empirically found that the space of growth rates

and lag times occupied by the vast majority of colonies is delimited by a front which is well described by approximating colony fitness as growth rate over lag time (Fig. 4A). Moreover, as hypothesized, colonies' growth rates and lag time are not uncoupled and hence are not uniformly distributed over the space of feasible growth parameters (Fig. 4B). Rather, most of the colonies operate in the proximity of a constant ratio between growth rate and lag time (Fig. 4B). Consistent with this front being Pareto optimal, we found that only in its proximity, growth rate exhibits a significant monotonic decrease with the increase of respiratory rate, in agreement with the theory of optimal proteome allocation[23] (Fig. 4B, C). Hence, we propose that fitness in a natural environment is largely invariant to fluctuations in respiratory activity, and as a consequence, expression noise in central metabolic enzymes had not been counter selected by evolution[3].

**Higher respiro- vs. fermentative metabolism prevents accumulation of toxic branched-chain amino acid intermediates.** The coexistence of diverse but equally optimal metabolic strategies that diversify cells towards fast growth or early growth

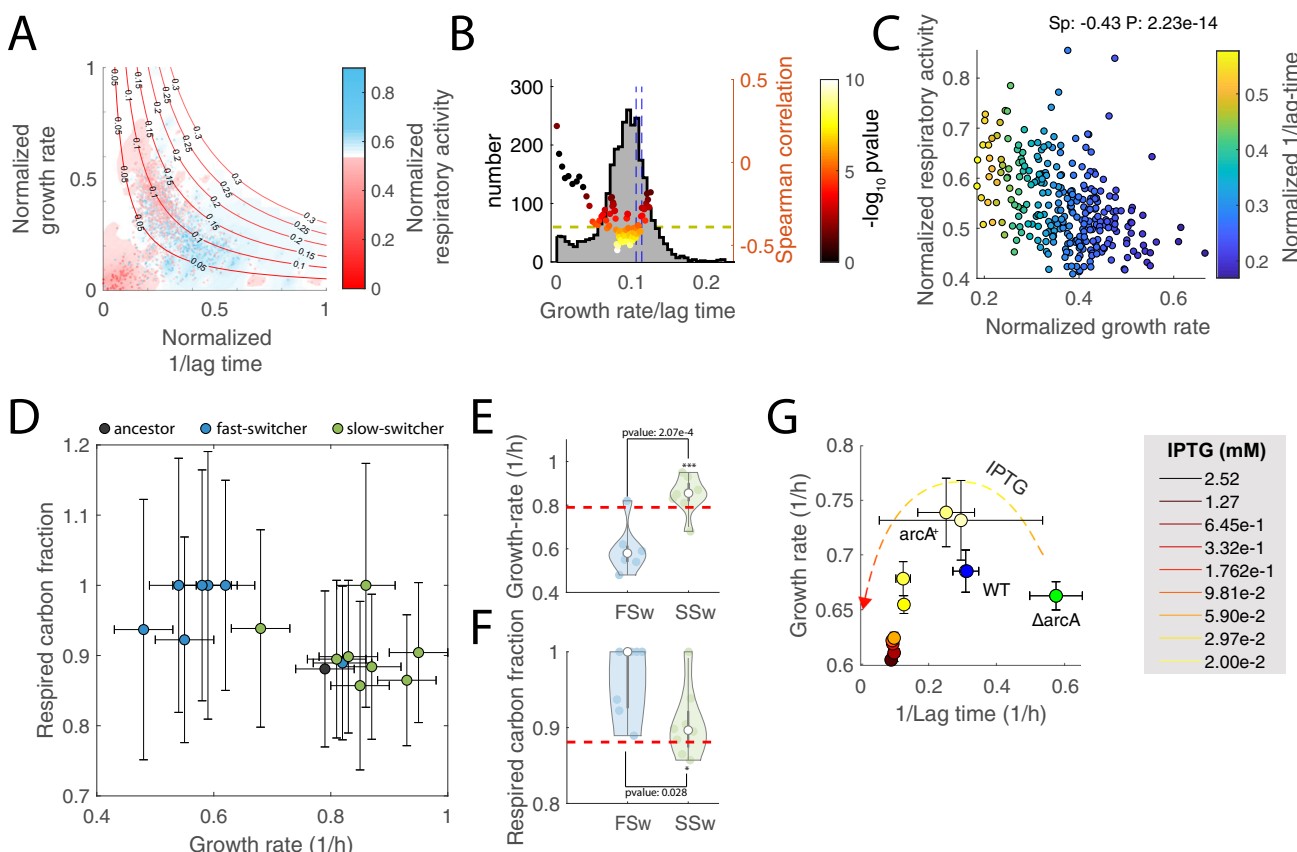

**Fig. 4 Respiro-/fermentative tradeoffs between growth rate and lag time. A** Growth rate and the inverse of the lag time is plotted for each colony of wild-type *E. coli* growing in three independent plates with glucose M9, same as Fig. 2C. Values have been normalized between 0 and 1. For each colony, the respiratory activity is color-coded (i.e., blue/red). The underlying heatmap reports on the respiratory rate estimated from averaging respiratory levels in proximal colonies. Isoclines identify points with a constant ratio between growth rate and lag time (i.e., approximation of colony fitness). **B** Histogram of the distribution of growth rate over lag time reported in panel (**A**). The green dashed line is the number of colonies expected by random choice in each histogram bin. Color-coded dots indicate Spearman correlation and significance (Spearman's Rho test) between growth and respiratory rates in colonies with increasing fitness levels (i.e., isoclines in panel (**A**)). Dashed blue lines highlight the 0.7 and 0.8 quantiles of the distribution. **C** Each dot reports on the respiratory activity and growth rate of a colony with a fitness value between blue dashed lines in panel (**B**). **D** Respiratory activity in FSw and SSw mutants is estimated as the fraction of glucose carbon that is not secreted as acetate, denoted as respired carbon fraction. Reported data are average ± standard deviation over three biological replicates. **E**, **F** Distributions of growth rates and respired carbon fractions in FSw and SSw strains. Red-dashed line reports on the ancestor level. Significance was calculated by a two-tailed *t*-test. **G** Lag time vs. growth rate in glucose minimal medium after 2 hours of carbon starvation in wild-type, ΔarcA and arcA+ mutants of *E. coli*. Reported data are average ± standard deviation over three biological replicates.

initiation is consistent with the evolutionary trajectories of *E. coli* evolved in batch liquid cultures containing glucose and acetate as sole carbon sources[43]. During adaptive diversification, two coexisting ecotypes emerge: one exhibiting fast growth in glucose but long lag time when switching from glucose to acetate, and one exhibiting shorter switching lag time and slow growth[43]. In addition to previous work demonstrating that fast-switchers (FSw) operate metabolism so as to require a minimum adjustment between growth on glucose and acetate[42], we predicted that FSw would favor respiratory metabolism, while slow-switchers (SSw) would feature higher fermentative metabolism (i.e., acetate overflow). To test our predictions we used previously published data[42] measuring growth rate, glucose-uptake, and acetate secretion for seven FSw and eight SSw randomly selected clones grown in liquid glucose minimal medium. In agreement with our predictions, we found that FSw on average have significantly ($p$ value = 0.028) higher respiratory activity and slower growth rates than SSw (Fig. 4D–F).

The evidence presented so far is mostly correlative. Hence, it is still unclear whether differences in respiratory activity are causal for changes in lag times, and whether they are sufficient to explain the tradeoff between growth and lag time. To test the causality of this relationship, we used a knockout strain of *arcA* (Δ*arcA*) with an isopropyl β-D-1-thiogalactopyranoside (IPTG) inducible promoter to modulate the *arcA* expression (*arcA*+). ArcA is a key repressor of genes involved in the TCA cycle and oxidative phosphorylation. While *arcA* deletion causes an increased carbon flux into the TCA cycle[44], respiration is inhibited and acetate fermentation increased upon *arcA* overexpression[45]. Hence, we expect lag times to increase with an increased expression of *arcA*. To test this hypothesis, we grew wild-type *E. coli*, the Δ*arcA* and *arcA*+ strains in batch liquid cultures of glucose minimal medium up to mid-exponential phase (optical density at 600 nm (OD$_{600}$) ~1), washed the cells, and incubated them in M9 medium without carbon sources together with the IPTG inducer (0.02 mM) for 2 h. Next, we resuspended cells in M9 medium with glucose and monitored OD$_{600}$ in a plate reader using different levels of IPTG inducer. Consistent with a previous study[45], mild induction of *arcA* (below 0.03 mM IPTG) is able to improve growth rate with respect to the wild-type (Fig. 4G). In addition, while Δ*arcA* grows slower than the wild-type, it also exhibits the shortest lag time, despite similar energy metabolism (Fig. S7). Overall, *arcA* expression levels directly correlate with the length of the lag. This phenomenon is similar when cells are resuspended in an M9 medium with a glycolytic (i.e., fructose) or gluconeogenic (i.e., acetate) substrate as sole carbon sources (Fig. S6). Altogether, this additional experimental evidence supports a key role of respiratory activity in facilitating rapid growth resumption and reveals the conflicting roles between respiration and fermentation, in preparing cells for rapid growth resumption and maximization of growth rates, respectively. Such a tradeoff might explain why at the population level *E. coli* cells are neither optimized for growth rate nor for lag time exclusively[42,45], but rather tend to adopt optimal tradeoffs, which may vary within the cell population (Fig. 4B, D).

Finally, we investigated how increasing respiro vs. fermentative metabolism in Δ*arcA* can affect the length of the lag period. To this end, we use metabolomics[46] to characterize the differences in the abundance of ~1000 metabolites between wild-type and Δ*arcA* in glucose M9 at exponential growth and during 2 h of carbon starvation (Supplementary Data 1). We found that exponentially growing Δ*arcA* mutants exhibit significantly ($q$ value < 1e−2) lower levels of biosynthetic intermediates in branched-chain amino acids (BCAA) (e.g., 2-methylmaleate, 2-isopropyl-maleate), including the main precursor pyruvate and the end product valine (Fig. 5A, B). On the other hand, dynamic metabolic changes upon carbon starvation are largely consistent between wild-type and Δ*arcA*, resulting in fewer and smaller metabolic differences, with the notable exception of non-degradable amino acids, such as valine and (iso-)leucine (Fig. 5C). Non-degradable amino acids accumulate during starvation[47] (Fig. 5D), but to a lower extent in Δ*arcA* than in the wild-type. Imbalance in the levels of BCAA intermediates can be highly toxic for *E. coli*, especially valine and leucine[48,49]. Intracellular accumulation of amino acids inhibits their own biosynthesis together with the biosynthesis of related amino acids[48] and the uptake of carbon sources[49–51]. Hence, we hypothesized that by maintaining lower levels of BCAA, Δ*arcA* is able to more rapidly resume the biosynthesis of biomass precursors once carbon becomes available. To test this hypothesis, we measured the lag time of carbon starved *E. coli* wild type and Δ*arcA* after supplementing glucose together with different amino acids: leucine, isoleucine, leucine+iso-leucine+valine, tryptophan, methionine, and glutamate (Fig. 5E, F). Δ*arcA* and wild-type exhibit a radically different response to supplementation of glucose and leucine after carbon starvation. While leucine sensibly prolongs lag time in wild-type, in Δ*arcA* leucine has a mild but beneficial effect (Fig. 5E, F). Moreover, BCAA supplementation in wild-type is able to reduce the lag-time by nearly 40%, more than costly amino acids[49], such as methionine (Fig. 5E). Altogether, experimental evidence suggests that the homeostasis of BCAA metabolism plays a crucial role during the initial growth of bacteria after starvation. We previously found that intracellular levels of the BCAA precursor pyruvate decrease with an increase in respiro vs. fermentative metabolism[49] (Fig. 5G). Metabolic changes induced by *arcA* deletion fit very well with the previously established relationship (Fig. 5G). We propose that by increasing respiratory capacity, Δ*arcA* has lower pyruvate levels (Fig. 5A) and consequently can maintain lower levels of BCAA intermediates (Fig. 5A), thereby reducing the risk of their toxic accumulation.

## Discussion

Here we monitored phenotypic and metabolic heterogeneity arising at a single colony level. A key advantage of this approach is the ability to directly monitor the variability in metabolic rates rather than reporting on the level of individual metabolites. It is worth noting that bacterial colonies develop in complex 3D structures in which individual bacteria can experience different metabolic gradients[52–54]. However, in *E. coli*, the first phase of radial colony growth is not limited by nutrient gradients[53], suggesting that fundamental characteristics of colony-forming cells, such as protein levels, can propagate to daughter cells and generate spatial correlations[54,55] detectable at the colony level. Moreover, transcriptional regulation of metabolism (e.g., Cra) often involves negative feedback loops in which metabolites can directly regulate transcription factors activity[56]. Such type of regulatory interactions can function as negative integral feedback and provide quasi-adaptation to small perturbations[57,58], thereby increasing the number of generations that are necessary for daughter cells to significantly diverge from the transcription factor activity of the ancestor. Therefore, while clearly different from single-cell measurements, by simultaneously monitoring thousands of colonies and their variation in growth kinetic parameters and respiratory rates, we could make experimentally testable predictions on the functional impact of metabolic diversity on phenotypic heterogeneity.

In this study, we showed that the regulation of key proteins involved in funneling carbon into the TCA cycle plays a fundamental role in the phenotypic heterogeneity of colonies growing on solid media. While not all regulatory solutions can

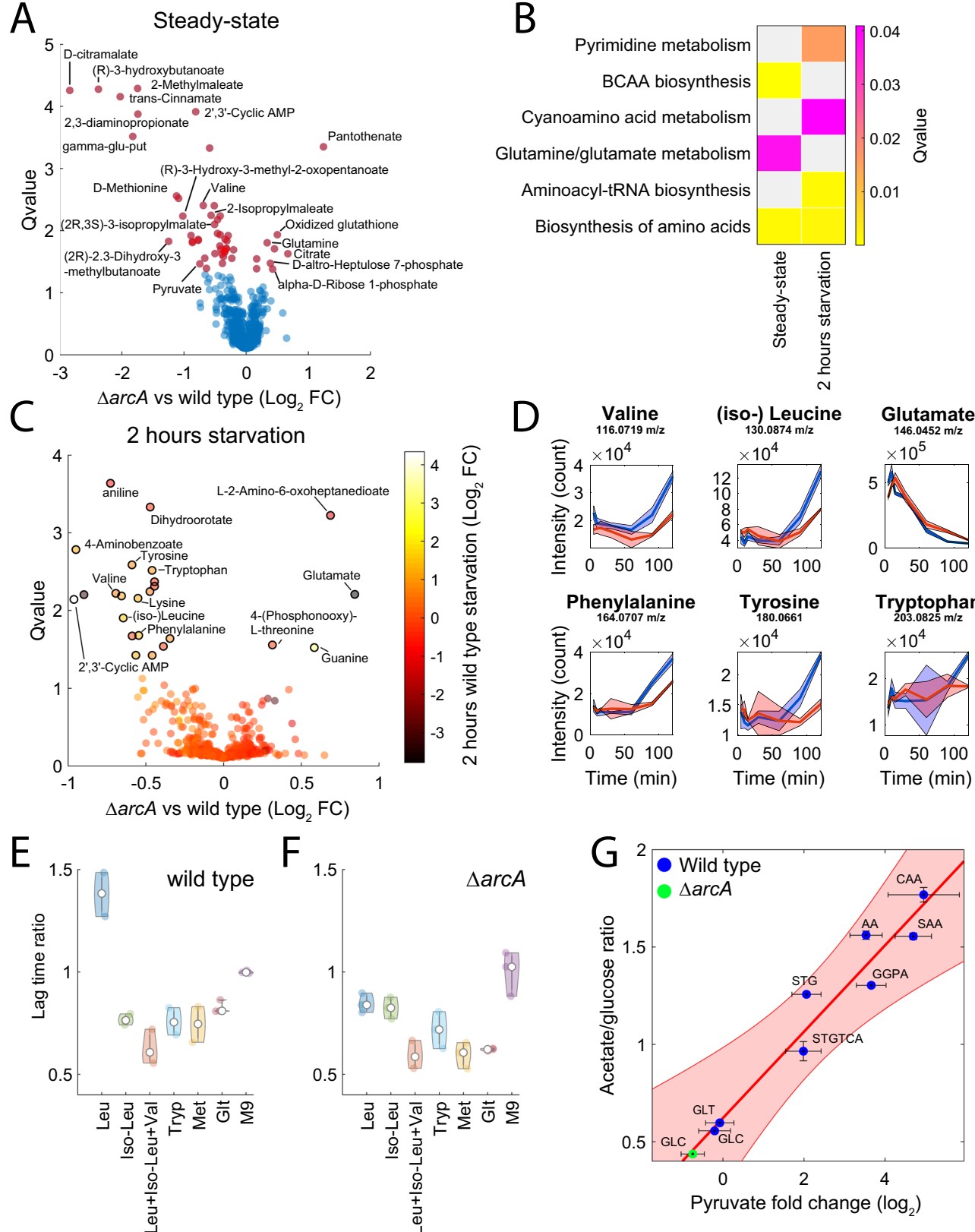

be equally optimal, the existence of multiple and largely diverse metabolic tradeoffs that are fitness-invariant can be a major driving force preserving and shaping phenotypic heterogeneity and plasticity—i.e., the ability to develop varied phenotypes

under fluctuating environmental conditions. Independent experimental evidence of planktonic cultures that evolve in coexisting subpopulations exhibiting the same predicted tradeoffs between lag time, growth rate, and respiratory activity,

**Fig. 5 Interplay between respiro-/fermentative metabolism and BCAA biosynthesis. A** Results from the metabolome-based analysis. Each dot in the volcano plot represents the relative difference in ion abundance for 955 putatively annotated metabolites between wild-type and $\Delta arcA$, averaged across three biological replicates. Metabolites with significant differences ($q$ value $\leq 0.05$, estimated by Storey correction of $p$ values from two-tailed $t$-test) are highlighted in red. **B** KEGG pathway enrichment analysis reporting metabolic pathways that are significantly ($q$ value $< 0.01$, estimated by Storey correction of $p$ values from permutation test) enriched for metabolites with a large difference between wild-type and $\Delta arcA$. **C** Volcano plot of metabolic differences between wild-type and $\Delta arcA$ after 2 h of carbon starvation (i.e., glucose deprivation), averaged across six biological replicates. For each metabolite, the color of the dot reflects the metabolite fold change in wild-type between 5 and 120 min of carbon starvation. Dots marked in black are metabolites that exhibit a significant ($q$ value $\leq 0.05$, estimated by Storey correction of $p$ values from two-tailed $t$-test) difference between wild-type and $\Delta arcA$. **D** Time course profiles of metabolite levels from 5 to 120 min of carbon starvation in wild-type (blue) and $\Delta arcA$ (red). We report the mean (thick line) ± standard deviation (shaded region) across six biological replicates. **E** Lag time of wild-type and **F** $\Delta arcA$, 2 h after carbon starvation. Cells were resuspended in M9 glucose with 1 mM of: leucine (Leu), isoleucine (iso-Leu), valine+leucine+iso-leucine (Leu + Iso-Leu+Val), tryptophan (Tryp), methionine (Met), and glutamate (Glt). We report the lag time of individual replicates relative to the average lag time in M9. **G** Difference in the abundance of pyruvate between $\Delta arcA$ (green dot—GLC) and wild-type in M9 glucose (panel (**A**)) against difference in respiro-/fermentative metabolism calculated as the ratio between acetate secretion and glucose uptake (mmol/gDW/h) estimated in[44]. Differences in $\Delta arcA$ were overlaid on previously acquired[49] relative changes in pyruvate levels and respiro-/fermentative metabolism in wild type across different nutritional environments: minimal medium with either glucose (GLC) or glucose minimal media supplemented with casamino acids (CAA), synthetic amino acid mix (SAA), SAA deprived of following amino acids: threonine, glycine, and serine (STG), threonine, glycine, serine, tryptophan, cysteine, and alanine (STGTCA), glutamate, glutamine, proline, and arginine (GGPA), aspartate and asparagine (AA), and glucose minimal medium with 0.125 g/L of glutamate (GLT). The red line represents the result from linear least squares regression analysis and 95% confidence intervals (shaded region). Reported data are average ± standard deviation over three biological replicates.

## Table 1 List of *E. coli* strains.

| Bacterial Strains | Collection | Reference |
|---|---|---|
| *E. coli* BW 25113 | KEIO collection | [79] |
| $\Delta$cra/$\Delta$arcA | KEIO collection | [79] |
| Cra/arcA overexpression plasmid | Obtained from the ASKA clone collection | [80] |

suggests that the same concepts can be generalized to largely diverse growth conditions.

Moreover, explaining colony-to-colony variations in growth rate and lag time by changes in respiro-/fermentative metabolism, together with the observation that an *arcA* deletion significantly suppresses resistance evolution[59], provides a basis to link independent experimental evidence relating lag time and energy metabolism[6,60,61] to antibiotic tolerance and resistance[7]. Consistent with our findings, mounting evidence associated evolutionary adaptation of bacterial persistence to mutations in enzymes of the respiratory metabolism, such as the proton-pumping NADH:ubiquinone oxidoreductase (Nuo complex)[61,62]. We demonstrated that homeostasis in BCAA can play a fundamental role during the early growth phases. We propose that valine/leucine and isoleucine and possibly intermediates in their biosynthesis can function similarly to toxin–antitoxin systems mediating bacterial dormancy. Toxin–antitoxin modules, such as HipAB[63], have well-established roles in regulating the bistability of clonal populations[64] and in the formation of growth-arrested persisters[63]. In a similar way, the accumulation of toxic BCAA intermediates in clonal cells with different metabolic activities can lead to cellular stasis by inhibiting BCAA biosynthesis and/or carbon uptake[49], ultimately affecting the biosynthesis of essential components for cell growth and division. The existence of different Pareto-optimal metabolic strategies within a clonal bacterial population, on the one hand, could represent a form of metabolic «division of labor»[65] and, on the other hand, could provide a bet-hedging strategy to withstand unforeseen challenges, such as periods of nutrient limitations or antibiotic treatments[6].

Overall, our findings can have important implications in diverse theoretical and applicative fields. Understanding the origin and functional impact of metabolic heterogeneity can open new opportunities in metabolic engineering and synthetic biology, to incorporate noise transmission in the design of more

efficient biosynthetic strategies. Moreover, fluctuations in respiro-/fermentative metabolism can be of particular significance in the design and interpretation of experimental evolution[7,8]. The existence of a tradeoff between shortest lag time and maximum growth rate and the ability to shift between these cellular objectives by modulating respiro-/fermentative metabolism can pave the way to alternative therapeutic strategies fighting the emergence of tolerant cells and eventually the appearance of drug resistance[66,67], potentially beyond antimicrobial treatments[68].

## Methods

**Strains and media**. For all growth experiments, *E. coli* BW25113 or mutants listed in Table 1 were initially grown overnight in LB or M9 minimal medium. LB medium consists per liter of deionized water of: 10 g Bacto-Tryptone (Becton Dickinson and Co.), 10 g NaCl, 5 g Yeast extract (DIFCO laboratories). The M9 medium contains per liter of deionized water: 7.5 g of $Na_2HPO_4$ $2H_2O$, 3.0 g $KH_2PO_4$, 1.5 g $(NH_4)_2SO_4$, and 0.5 g NaCl and was adjusted to pH 7 before autoclaving. The following components were filter-sterilized separately and then added (per liter of final medium): 1 mL of 1 M $MgSO_4$, 1 mL of 0.1 M $CaCl_2$, 1 mL 0.1 M $FeCl_3$, and 10 mL of a trace element solution containing (per liter) 180 mg $ZnSO_4$ $7H_2O$, 120 mg $MnSO_4$ $H_2O$, 180 mg $CoCl_2$ $6H_2O$, and 120 mg $CuCl_2$ $2H_2O$. Carbon source solutions were filter-sterilized and added separately to the medium, 5 g/L glucose, fructose, acetate. Agar plates were made by adding 15 g/L of Agar.

Strains used for the experiments discussed in the main text, their origin, and corresponding reference.

**Growth rate measurements**. The growth rate and lag time in batch liquid culture were measured by monitoring $OD_{600}$ in 96-well plates with the multiwell reader Infinite (Tecan, Switzerland) at 37 °C with shaking. The growth rates were extracted by fitting the exponential part of the growth. An iterative approach was used to find the time window of at least 100 min with the maximum exponential growth rate. The intersection between the initial OD of the inoculum and the extension of the exponential growth curve yielded the lag time[69].

**Statistical analysis**. Statistical analyses were performed using Matlab R2018b (MathWorks). Gene ontology and metabolic pathway enrichment analysis, and enrichment of noisy enzymes in TF-target genes were based on an iterative hypergeometric test described in Ref. [70]. When necessary, $p$ values were corrected for multiple tests by $q$ value estimation[71]. The network of TF-target genes reported in[72] was used to identify TF regulating proteins with large deviating noise.

**Image acquisition and analysis**. Overnight cultures were diluted in LB or M9 media 1:1000. Cells were grown up to an OD of 1 before plating at appropriate dilutions on solid agar medium. Standard PC scanners (i.e., Epson Perfection V370 Photo) were used to acquire images at 800 dpi resolution of Petri dishes every 10 min. Acquired images were analyzed with an in-house image analysis software implemented in ImageJ IJ1 Macro and Python. In order to improve the accuracy in detection of bacterial colonies the software corrects for changes in light across plates (i.e., Otsu thresholding over each plate), performs colony segmentation (i.e., watershed algorithm over the standard variation of the experiment's Z-stack

projection of all images) and tracking. Finally, outliers/artifacts are filtered out. The final data returns for each colony $x$ and $y$ coordinate on the plate, size at each time point, and RGB intensity. Full details and computer codes are available on GitHub https://github.com/Dfernand1795/PetriScanner2.

**Estimation of deviating noise**. Here we used time-independent protein ($p$) abundance and noise from single-cells of *E. coli* from[13]. Consistent with other studies, at low copy number ($\mu_p$) the noise ($\eta_p = \sigma_p^2/\mu_p^2$, where $\sigma_p$ is the standard deviation) is dominated by intrinsic noise (e.g., stochastic effects in gene expression), while at high expression levels noise is dominated by extrinsic factors, such as fluctuations in ribosomes, polymerase, transcription factors and partitioning of the population between the cell-cycle stages[73–75]. To estimate average noise levels at different protein abundance, we used a moving average smoothing method on the data reported for 1018 proteins[13]. Specifically, we used the *maloweess* function in Matlab, where for each protein, the expected average noise level ($\bar{\eta}_p$) is estimated from the average of the 5% closest proteins:

$$\bar{\eta}_p = \frac{\sum\limits_{\Omega = lb_p \le \mu_i \le ub_p} \eta_i}{|\Omega|} \tag{1}$$

Deviating noise is calculated as the ratio between the noise of individual proteins and average noise levels of proteins with similar abundance: $\varepsilon_p = \eta_p/\bar{\eta}_p$.

**Estimation of colony maximum growth rate, lag time, and respiratory rate**. For each colony, the sigmoidal changes in the area over time were fitted by a Gompertz function[76]

$$f = A_{\min} + \frac{(A_{\max} - A_{\min})}{e^{e^{k(t_m - t)}}} = A_{\min} + \frac{A_\Delta}{e^{e^{k(t_m - t)}}}$$
$$f' = A_\Delta e^{-e^{k(t_m - t)}} \cdot k e^{k(t_m - t)}$$
$$f'' = A_\Delta k^2 e^{-e^{k(t_m - t)}} \cdot e^{k(t_m - t)}(e^{k(t_m - t)} - 1)$$
$$f'_{(t = t_m)} = \frac{A_\Delta}{e} k \tag{2}$$
$$T_{lag} = \frac{A_{\min} + \frac{A_\Delta}{e}(1 - k t_m)}{\frac{A_\Delta}{e} k}$$

The Gompertz function is able to describe asymmetrical growth curves assuming that at the time of inflection $t_m$ the maximum growth rate is achieved. Parameter $k$ represents the maximum relative growth rate. $A_{max}$ and $A_{min}$ the maximum and minimum area respectively. For simplicity, we assume colony size at inoculation to be equal to zero. By fitting this function to each colony growth curve we can analytically estimate maximum growth rate ($f'$) and lag-time ($T_{lag}$). Fitting was performed in Matlab 2018b using the *lsqcurvefit* function using the Truest-region algorithm for optimization from 50 multiple start points (i.e., *MultiStart* function). To determine the maximum respiratory rate we use a linear model and found the time interval (consisting of at least ten time points) with the largest proportional coefficient between time and average red intensity of a colony. $f = \alpha t + \beta$, where $\alpha$ is the estimate for maximum respiratory rate (AU/h) and $\beta$ is an offset value.

**ATP reporter assay**. The low concentration ATP reporter plasmid pRS-QUE7mu[21] was transformed into *E. coli* bacterial strains BW25113 and BW25113-ΔarcA (Keio:JW4364) expressing a T7 RNA polymerase (araB::T7RNAP-tetA). For each sample, a single bacterial colony was grown in LB medium for 6 h at 37 °C under agitation (170 RPM). 30 microliters of culture was pelleted, resuspended in 3 ml of M9 medium and further diluted 1:10 in 3 ml of M9 medium supplemented with Glucose 0.5%, ampicillin 50 µg/ml and arabinose 0.01%. Cultures were grown for 16 h at 37 °C under agitation (170 RPM) and typically reached OD600 values between 0.05 and 0.3. For the starvation procedure, cells were washed twice and resuspended in 3 mL of M9 medium only supplemented with ampicillin 50 µg/ml. Samples were brought back to the shaker for two hours before addition of glucose at 0.5%. Cultures were sampled for FACS analysis right before the starvation phase, after 30 and 120 min of starvation and after one and 15 min after addition of glucose. 30 µl of sampled culture was diluted in PBS containing propidium iodide (1 µg/ml, ThermoFisher:P3566). FACS measurements were performed on a BD FACSaria III Cell sorter. Fluorescence was measured with the following channels Ex488_LP495_BP514/30-H, Ex405_LP502_BP530/30-H and Ex488_LP610_BP616/23-H (for viability; propidium iodide).

**Metabolome profiling**. Wild-type and *ΔarcA E. coli* overnight cultures growing on M9 minimal medium were diluted in fresh M9 glucose minimal media and grown at 37 °C until exponential phase and an $OD_{600}$ of 1. Cells were washed twice with M9 medium without carbon source and 700 µL cell cultures were distributed in 96 deep well plates and incubated at 37 °C until shaking at 2 RCF. Samples for metabolomics profiling were taken during exponential growth prior to carbon deprivation and 5, 10, 15, 30, 60, 90 and 120 min after carbon starvation. In total, 50 µl of whole cell broth was directly transferred to 150 µl extraction liquid solution containing 50% (v/v) methanol and 50% (v/v) acetonitrile at −20 °C. The extraction was carried out by incubating the samples for 1 h at −20 °C. Samples were centrifuged for 5 min at 1789

RCFand 80 µl of the supernatant was transferred to 96 well storage plates and stored at −80 °C (Supplementary Data 1). The mass spectrometry analysis was performed on a platform consisting of an Agilent Series 1100 LC pump coupled to a Gerstel MPS2 autosampler and an Agilent 6550 Series Quadrupole Time of Flight mass spectrometer (Agilent, Santa Clara, CA) following the protocol described in[77]. Mass spectra were recorded from m/z 50 to 1000 using the highest resolving power (4 GHz HiRes). All steps of mass spectrometry data processing and analysis were performed with MATLAB (The Mathworks, Natick). Detected ions were matched to a list of metabolites based on the corresponding molar mass[78]. For the full list of metabolites used for annotation see Supplementary Data 1.

**Reporting summary**. Further information on research design is available in the Nature Research Reporting Summary linked to this article.

## Data availability
All data generated or analyzed during this study are included in this published article as Supplementary Data 1. A detailed description of all data analysis steps is published in this article in the Methods section and Supplementary Information. Source data are provided with this paper.

## Code availability
Python code for the image analysis software is available for download at http://www.imsb.ethz.ch/research/zampieri-group/resources.html and https://github.com/Dfernand1795/PetriScanner2.

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

## Acknowledgements

We thank Uwe Sauer for supporting this work and providing laboratory facilities, Elad Noor, Maren Diether, Wolfram Liebermeister, Karin Ortmayr and Roberto de la Cruz Moreno for helpful feedback and discussions, the Balaban group for initial discussion on the scanner array setup. This project was supported by the National Center of Competence in Research AntiResist funded by the Swiss National Science Foundation (grant number 51NF40_180541).

## Author contributions

M.Z. designed the project. M.Z. and D.F. performed the experiments and analyzed the data. D.F. implemented the image analysis framework. P.M. and U.J. design and implemented ATP single-cell measurements. All authors contributed to preparing the manuscript.

## Competing interests

The authors declare no competing interests.
