## [Peer Review File · Nature Communications]

Reviewers' Comments:

Reviewer #1:

Remarks to the Author:

General comments

The authors of the study "Pareto optimality couples cellular noise to phenotypic heterogeneity" developed a high-throughput assay that uses a redox-sensitive dye to couple growth of bacterial to their respiratory activity. They could show that in *Escherichia coli*, noise of the lower glycolysis and citric acid cycle is responsible for large variations in respiratory metabolism. They found that these variations are Pareto optimal to maximization of growth rate and minimization of lag time. The manuscript is well written, experiments were carefully planned, performed and analysed. The manuscript contains a logical structure and is well organized; hypotheses are experimentally verified and documented. Statistical analysis was performed where necessary and I am convinced with the information given in the material and method section as well as in the supplement material the ability of a researcher to reproduce the work is given.

The presented data are novel and could be of interest for a wide range of readers, ranging from biochemistry, microbiology to systems biology.

Before publication, there are some points and comments to be clarified and addressed.

Major point:

- 1.) Title: Can the authors comment, why they choose the existing title, rather than e.g. "Pareto optimality couples growth rate and lag time". Furthermore, the principle was only shown for *E. coli*. This should become clear in the title.
- 2.) Experiments: Although the authors investigated 1000s of colonies, in my opinion it is not clear if the experiments lack replication. Can you comment on independent biological replicates (independent cultures grown and analysed in the same way)? Three are needed! If this is not given, the lack of replication precludes publication, since it does not provide solid biological insights.

Minor points:

Page 2, bottom: Delete one "that"

Page 3, top: "can be fitness invariant". Please clarify why "can" and not "is"?

Page 7: Can you specify to which specific supplement material you are referring to?

Figure S2: change "an" to "and"

Figure S3: *E. coli* not italic

Figure S3C: Grey cross not visible

Figure S3D: Parts of the box are not visualized. Image quality needs improvement

Reviewer #2:

Remarks to the Author:

In this manuscript Fernandez-Fuentes et al. analyzed previously published single cell proteome data of *E. coli*, and estimate deviating noise of each protein, i.e. how much expression of a protein deviates from proteins in the same expression range. They concluded that expression noise leads to difference in protein abundances, which may result in cell-to-cell variability of metabolism and consequently in phenotypic heterogeneity. They then show that especially respiration-/fermentative metabolism should have a high cell-to-cell variability, because these enzymes are expressed at critical levels and show high deviating noise.

To test (single cell) heterogeneity of respiration-/fermentative, the author's developed an innovative multiparametric assay in agar plates coupled with a colorimetric change (TTC to TFP), to monitor

growth rates (as a function of time and colony size -area-), lag times and respiratory rates (as a function of time and color change intensity) in *E. coli* cells. The assay showed a negative correlation between lag-times and respiratory activity, which lead to the hypothesis that respiratory activity are linked to lag-times. The authors further propose that the noise observed in central metabolism enzymes is a phenomenon that has not been selected out because there is a risk-benefit trade off between respiratory activity and lag-time. In this way, in a population there is a fitness-neutral metabolism, where different strategies are employed by all the members of the population, and each "pareto optimal" state establishes a compromise between growth rate and lag time. The paper is very interesting, nice to read and puts forward new hypothesis about metabolic heterogeneity. However, my major concern is that the colony measurements do not allow conclusions about phenotypic heterogeneity of an individual cell, they rather show heterogeneity of a population.

Major Points

1. How can measurements of bacterial colonies inform about cell-to-cell variation of metabolism? Bacterial colonies are arguably complex biofilm-like communities, where single cells experience different nutritional conditions depending on their position in the colony. This affect especially respiro-/fermentative because oxygen is abundant at the surface but not inside the colony. Therefore it is not clear to me how data from the multiparametric assay with bacterial colonies can inform about phenotypic heterogeneity and cell-to cell variation of metabolism? I can imagine that lag-times are related to the metabolic state of the first cell in the colony, but I don't understand how growth rates and respiratory activity of the colony are related to metabolism of a single cell. Instead, it has been shown that the conditions in bacterial colonies can lead to subpopulations within the colony and to a complex collective behavior (e.g. <https://doi.org/10.1186/s12918-015-0155-1>). I suggest that the authors at least explain and discuss the assumptions that underlie their conclusion about single cell effects and heterogeneity of single cells vs heterogeneity of the population in a single colony.
2. The colorimetric assay (TTC to TFP) is very interesting but the authors should better validate the method. They could test (with bulk measurements) that the dye does not affect physiology, especially metabolism, and use conditions that differ in respiratory rates to calibrate the assay.
3. The hypothesis about pareto optimality is compelling and probably true. But I have problems to understand the results in Figure 4A. Is the relationship significant? Further, the authors should discuss under which conditions the pareto optimality occurs, only in biofilm cultures or in planktonic cultures as well.

Minor points

Figure 1 – In the legend, change E) to F)

Figure 3A – Remove drawing of dropper, it's not needed

Figure S3C – Change the size or color of the gray cross (maybe use a black dot?), it's difficult to see it in the printed and digital versions

Figure S7 – the x-axis is unclear

Reviewer #3:

Remarks to the Author:

Fuentes et al present their work on growth and metabolic heterogeneity at the level of single colonies of *E. coli*. They start by observing from previously published data that high deviating expression noise, i.e. expression noise of a protein relative to the noise of all proteins with comparable mean expression, is enriched for metabolic genes involved in oxidative-reductive reactions. The authors postulate that selective pressure may favour high noise levels in these enzymes as it can translate to phenotypic heterogeneity in growth and metabolic activity, which may be beneficial in fluctuating environments. They measure respiratory rate of colonies and examine correlations among respiratory rate, maximal growth and lag-time, finding strongest correlation between lag-time and respiratory

rate, which they confirm to hold true for a variety of growth conditions. They go on to argue that respiratory heterogeneity explains phenotypes wrt growth rate and lag time, which they find confined to regions delimited by a Pareto-front. The authors therefore conclude that cells may face a trade-off between two competing objectives, namely maximisation of growth rate and minimisation of lag-time. By deleting/over-expressing a key repressor of respiratory enzymes, they further trace back the correlation between respiratory activity and lag-time to the build up of some intermediates in branched chain amino acids. Finally they conclude that such build up could lead to toxic levels which may explain slow growth resumption, i.e. long lag, and discuss links and implications for antibiotic tolerance associated with same said repressor.

I only have minor comments, see below. In particular, I feel that some of the figures could be improved to increase overall quality of presentation and general readability. Altogether, the article is well written, coherent, and while I cannot comment on the details of the experimental protocols, in my view experiments support the conclusions drawn by the authors. The work should be interesting to a broad readership - I would recommend this paper for publication in Nature Communications after careful revision of the comments below.

Figure 1:

A - Left axis (standard) wrongly labelled? Should be log. A different choice of colours would help distinguish points for standard and deviating noise. In general, I think the double axis introduces more confusion than necessary, especially since they are practically the same scale. Instead the legend should better indicate which markers refer to which type of noise, e.g. green dots unclear.

B & C - There is no reference to asterisks - do they refer to significance levels? Connected to that, it's also unclear to me why authors conclude from this in the main text that essential genes are under selective pressure for noise reduction, as deviating noise is comparable to that of all genes.

E - What does the size of circles refer to, # target genes? Is that what the blob next to it is supposed to say? The caption should be more explanatory.

F - Wrongly referenced in caption by 'E'. Consider different colours avoiding red/green (also in Fig. S2). Some of the regression lines are solid instead of dashed - any meaning?

Figure S1B Caption:

Consider referring to 'standard/deviating' rather than 'absolute/relative' noise for consistency.

Figure 2:

A-E - I find the ordering of plots confusing. If it's meant as some sort of matrix view then x- and y-axis should be given by rows and columns, but this is not the case. This makes it harder to figure out, which plots to compare. Also the scaling of comparable axes differs, which makes it hard to compare trends between LB and M9. For example, could it be that there are two different growth rate regimes where correlation with the other variables is opposed. Case in point, correlations in Fig. 3D-F, where growth rate has been reduced, are more similar to the lowly-sampled regions in Fig. 2. These opposing trends also explain the poor correlation of growth with other variables, which is later investigated as significant in the Pareto analysis. Consider increasing some fonts, e.g. Sp & P. Why are there different units for respiratory activity in the y-labels?

H - When referred to from main text, what does K refer to?

Figure 4:

A - In caption, what does 'black dots' refer to?

D - In caption, 'mutants' mis-spelled.

Figure 5:

C - The colour map doesn't cover all colours shown, e.g. what do purple/blue dots mean? Same x-/y-scaling as in A would help conveying the argument that after starvation we see fewer change compared to wt.

D - Again, preserving scaling across plots would help comparison.

Figure S3:

C - red line is invisible.

E&F - Re-ordering the legend and considering similar colour for +/-TCC would make the case more obvious.

Figure S4:

A-C - What are the numbers in the legend? CVs? Caption could be more explanatory.

Figure S5:

The caption reads a bit confusing, e.g. 'scatter of individual conditions (red dots) overlaid on overall density' could be clearer.

Figure S7:

What happened to the x-label?

Typos:

- Introduction: "Surprisingly, we found that that individual colonies..."

- After reference to Fig. 2A-D: "Coefficient of Variance" → "Coefficient of Variation"

Other:

- Personally, I feel that 'SS' as abbreviation for slow switchers seems unfortunate historically and could be avoided.

Reviewed by Andrea Weisse, U Edinburgh

Reviewer's Comments:

Reviewer #1 (Remarks to the Author)

General comments

The authors of the study "Pareto optimality couples cellular noise to phenotypic heterogeneity" developed a high-throughput assay that uses a redox-sensitive dye to couple growth of bacterial to their respiratory activity. They could show that in *Escherichia coli*, noise of the lower glycolysis and citric acid cycle is responsible for large variations in respiratory metabolism. They found that these variations are Pareto optimal to maximization of growth rate and minimization of lag time. The manuscript is well written, experiments were carefully planned, performed and analysed. The manuscript contains a logical structure and is well organized; hypotheses are experimentally verified and documented. Statistical analysis was performed where necessary and I am convinced with the information given in the material and method section as well as in the supplement material the ability of a researcher to reproduce the work is given.

The presented data are novel and could be of interest for a wide range of readers, ranging from biochemistry, microbiology to systems biology.

We thank the reviewer for the encouraging and positive comments and helpful feedbacks.

Before publication, there are some points and comments to be clarified and addressed.

Major point:

1.) Title: Can the authors comment, why they choose the existing title, rather than e.g. "Pareto optimality couples growth rate and lag time".

While the tradeoff between lag time and growth rate is a major finding of our work, in our view the emphasis should also be on how optimal tradeoffs can shape cellular noise. We have now changed the title: Pareto optimality between growth rate and lag time couples metabolic noise to phenotypic heterogeneity in *Escherichia coli*.

Furthermore, the principle was only shown for *E. coli*. This should become clear in the title.

We now made this clear in the new title

2.) Experiments: Although the authors investigated 1000s of colonies, in my opinion it is not clear if the experiments lack replication. Can you comment on independent biological replicates (independent cultures grown and analysed in the same way)? Three are needed! If this is not given, the lack of replication precludes publication, since it does not provide solid biological insights.

We thank the reviewer for raising this important point, which we now clarify also in the figure captions. Experiments (including those performed in liquid media) were done in triplicates. For the perturbation analysis reported in Fig 3, 14 different plates corresponding to the 14 different conditions (1 control + 13 chemical perturbations) were measured independently and pulled all together. In Fig. S5 we reported the independent results for each condition showing that in all of them we found the same strong anticorrelation between lag time and respiratory activity. The number of colonies per dish, including replicates, analyzed in all experiments of this study is

reported in the Supplementary Data S1. Fig. S3 panel E and F reports the distribution obtained from 3 independent plate replicates, showing the reproducibility of experiments. We revised the figure to improve clarity.

Minor points:

Thanks for the suggestions and spotting the mistakes.

Page 2, bottom: Delete one “that”

OK

Page 3, top: “can be fitness invariant”. Please clarify why “can” and not “is”?

Not all possible ways of regulating respiration-/fermentative metabolism can achieve an optimal tradeoff between lag time and growth rate (i.e. being Pareto optimal). We now explicitly mention this aspect in the discussion of the manuscript.

Page 7: Can you specify to which specific supplement material you are referring to?

We now clarify that we referred to the supplementary text included in the supplementary material pdf: “Lag time vs growth rate”.

Figure S2: change “an” to “and”

OK

Figure S3: E. coli not italic

OK

Figure S3C: Grey cross not visible

OK

Figure S3D: Parts of the box are not visualized. Image quality needs improvement

OK

Reviewer #2 (Remarks to the Author)

In this manuscript Fernandez-Fuentes et al. analyzed previously published single cell proteome data of E. coli, and estimate deviating noise of each protein, i.e. how much expression of a protein deviates from proteins in the same expression range. They concluded that expression noise leads to difference in protein abundances, which may result in cell-to-cell variability of metabolism and consequently in phenotypic heterogeneity. They then show that especially respiration-/fermentative metabolism should have a high cell-to-cell variability, because these enzymes are expressed at critical levels and show high deviating noise.

To test (single cell) heterogeneity of respiration-/fermentative, the author’s developed an innovative multiparametric assay in agar plates coupled with a colorimetric change (TTC to TFP), to monitor growth rates (as a function of time and colony size -area-), lag times and respiratory rates (as a function of time and color change intensity) in E. coli cells. The assay showed a negative correlation between lag-times and respiratory activity, which lead to the hypothesis that respiratory activity are linked to lag-times. The authors further propose that the noise observed in central metabolism enzymes is a phenomenon that has not been selected out because there is a risk-benefit trade off

between respiratory activity and lag-time. In this way, in a population there is a fitness-neutral metabolism, where different strategies are employed by all the members of the population, and each “pareto optimal” state establishes a compromise between growth rate and lag time. The paper is very interesting, nice to read and puts forward new hypothesis about metabolic heterogeneity. However, my major concern is that the colony measurements do not allow conclusions about phenotypic heterogeneity of an individual cell, they rather show heterogeneity of a population.

We thank the reviewer for the overall positive comments and for very insightful comments that helped to mature the paper and that we now included in a revised version of the manuscript. Specifically, we revised the text to avoid any overstatement and clarify the difference between colony and single cell variability.

Major Points

1. How can measurements of bacterial colonies inform about cell-to-cell variation of metabolism? Bacterial colonies are arguably complex biofilm-like communities, where single cells experience different nutritional conditions depending on their position in the colony. This affect especially respiro-/fermentative because oxygen is abundant at the surface but not inside the colony. Therefore it is not clear to me how data from the multiparametric assay with bacterial colonies can inform about phenotypic heterogeneity and cell-to cell variation of metabolism? I can imagine that lag-times are related to the metabolic state of the first cell in the colony, but I don't understand how growth rates and respiratory activity of the colony are related to metabolism of a single cell. Instead, it has been shown that the conditions in bacterial colonies can lead to subpopulations within the colony and to a complex collective behavior (e.g. <https://doi.org/10.1186/s12918-015-0155-1>). I suggest that the authors at least explain and discuss the assumptions that underlie their conclusion about single cell effects and heterogeneity of single cells vs heterogeneity of the population in a single colony.

The reviewer raised a very important point which we did not adequately addressed.

We fully agree with the reviewer that colonies develop in complex 3D structures that consists of communities in which individual bacteria can experience different metabolic gradients (e.g. oxygen availability)^{1 2 3}. However, estimated parameters like maximum growth rate or respiratory activity associate to the very first phases of colony growth (see also new Fig S3 G), during which radial growth dominates and has been shown not to be limited but nutrient gradients². During such phase, colony growth can be approximated as mostly radial and hence suggests that (at least at the beginning) colony structure is less complex and possibly individual cells experience similar conditions.

Moreover, there are multiple mechanisms by which characteristics of the first cell in a colony can propagate to “daughter” cells, and hence dominate the average colony behavior in the early growth phase, before these characteristics are lost.

In *E. coli* most proteins are not actively degraded. Therefore, higher/lower levels of proteins are likely to propagate to daughter cells, generating spatial correlations and leading to neighbouring cells in bacterial groups to have similar gene expression levels³. This holds true also for other more complex organisms like mammalian cells⁴.

Another key element that could explain why (especially) regulation of respiration/fermentative metabolism can robustly propagate in cell lineages is the direct regulatory link between metabolites and transcription factor activity. For example, Cra activity was shown to be allosterically inhibited by FBP⁵. Such type of regulatory interactions can function as integral regulatory feedbacks and provide (quasi) perfect adaptation to small perturbations⁶. As a direct consequence, feedback regulatory mechanisms can increase the number of generations that are needed to significantly diverge from the ancestor in the activity of transcription factors (e.g. Cra).

We have now revised the text to avoid any overstatements. We clarified that heterogeneity measured with our assay is derived from colony-to-colony variations and discuss the key differences with respect to single cell studies in planktonic culture. We also discuss the assumptions needed to derive analogies between single cells and colony-to-colony variability including the suggested reference. It is also worth noting that predictions of a tradeoff between lag time and growth rates mediated by respiration/fermentative metabolism (that we developed on the basis of the colony assay) was tested in planktonic cultures (see point4).

2. The colorimetric assay (TTC to TFP) is very interesting but the authors should better validate the method. They could test (with bulk measurements) that the dye does not affect physiology, especially metabolism, and use conditions that differ in respiratory rates to calibrate the assay.

Unfortunately, it is unavoidable that TTC will interfere with metabolism. TTC is reduced mostly by NADH, sequestering it as a reducing agent in oxidative phosphorylation. The effects are shown in Fig. S3 (now revised for clarity). However, we showed (Fig. S3 panel E and F) that it doesn't significantly affect colony to colony variations. We now clarify this aspect in the main text. We could also show that consistent with independent measurements of respiratory rates in liquid media⁷, estimates of respiratory activity in complex media is higher than in minimal media (Fig. 2E).

3. The hypothesis about pareto optimality is compelling and probably true. But I have problems to understand the results in Figure 4A. Is the relationship significant?

We did not adequately describe this analysis. We revised the text to clarify the analysis and results reported in Fig. 4A. The significance of the relationship found in Fig. 4A is investigated in Fig. 4B, where we projected the data points along growth-rate over lag-time (e.g. the isoclines reported in Fig 4A). The distribution shows that growth rates and lag times of colonies are not random, and hence don't distribute uniformly (Green dashed line in Fig. 4B). Rather, most colonies approximated a fixed ratio (proximal to 0.1). Colonies proximal to this tradeoff between lag time and growth rate exhibited the most significant anticorrelation between growth rate and respiratory activity (dot points in Fig. 4B, see also Fig. 4C). Such anticorrelation was shown to underly optimal growth strategies in *E. coli*⁸, supporting that estimated tradeoffs are optimal (i.e. Pareto front).

Further, the authors should discuss under which conditions the pareto optimality occurs, only in biofilm cultures or in planktonic cultures as well.

The reviewer has a point. While we don't directly report single cell data of planktonic culture, the same tradeoff found in colonies growing on solid agar plates is shown to hold true also when measuring population averaged parameters (e.g. lag time/growth rate/respiratory rate) in planktonic cultures. Data presented in Fig. 4D,E,F,G are derived from experiments performed in liquid media (shake flasks). We revised the text to avoid any overstatement, and clarify this point in the discussion. The presented data demonstrated: i) the coexistence of diverse evolved

subpopulations that exhibit fast growth or rapid switch and that, consistently with our predictions, have lower or higher respiratory activity, respectively, ii) modulating the expression of *arcA* is sufficient to change respiration/fermentative metabolism and consequently change the tradeoff between growth-rate and lag time (as predicted from the colony assay). Specifically, deletion of *arcA* (e.g. more respiration) favors shorter lag time and slower growth rate, while mild overexpression can increase growth rate at the cost of longer lag time. It is worth noting that also all data reported in Fig. 5 were generated from *E. coli* cultivated in shake flasks.

Minor points

Thanks for the suggestions and spotting the mistakes.

Figure 1 – In the legend, change E) to F)

OK

Figure 3A – Remove drawing of dropper, it's not needed

OK

Figure S3C – Change the size or color of the gray cross (maybe use a black dot?), it's difficult to see it in the printed and digital versions

OK

Figure S7 – the x-axis is unclear

OK

Reviewer #3 (Expertise: Experimental and computational modeling of stochasticity in bacterial cell behavior):

Fuentes et al present their work on growth and metabolic heterogeneity at the level of single colonies of *E. coli*. They start by observing from previously published data that high deviating expression noise, i.e. expression noise of a protein relative to the noise of all proteins with comparable mean expression, is enriched for metabolic genes involved in oxidative-reductive reactions. The authors postulate that selective pressure may favour high noise levels in these enzymes as it can translate to phenotypic heterogeneity in growth and metabolic activity, which may be beneficial in fluctuating environments. They measure respiratory rate of colonies and examine correlations among respiratory rate, maximal growth and lag-time, finding strongest correlation between lag-time and respiratory rate, which they confirm to hold true for a variety of growth conditions. They go on to argue that respiratory heterogeneity explains phenotypes wrt growth rate and lag time, which they find

confined to regions delimited by a Pareto-front. The authors therefore conclude that cells may face a trade-off between two competing objectives, namely maximisation of growth rate and minimisation of lag-time. By deleting/over-expressing a key repressor of respiratory enzymes, they further trace back the correlation between respiratory activity and lag-time to the build up of some intermediates in branched chain amino acids. Finally they conclude that such build up could lead to toxic levels

which may explain slow growth resumption, i.e. long lag, and discuss links and implications for antibiotic tolerance associated with same said repressor.

I only have minor comments, see below. In particular, I feel that some of the figures could be improved to increase overall quality of presentation and general readability. Altogether, the article is well written, coherent, and while I cannot comment on the details of the experimental protocols, in my view experiments support the conclusions drawn by the authors. The work should be interesting to a broad readership - I would recommend this paper for publication in Nature Communications after careful revision of the comments below.

We thank the reviewer for the positive and constructive comments which help to improve clarity.

Figure 1:

A - Left axis (standard) wrongly labelled? Should be log. A different choice of colours would help distinguish points for standard and deviating noise. In general, I think the double axis introduces more confusion than necessary, especially since they are practically the same scale. Instead the legend should better indicate which markers refer to which type of noise, e.g. green dots unclear.

B & C - There is no reference to asterisks - do they refer to significance levels? Connected to that, it's also unclear to me why authors conclude from this in the main text that essential genes are under selective pressure for noise reduction, as deviating noise is comparable to that of all genes.

Thanks for the helpful suggestions. We implemented the suggested changes, and clarified the significance associated to asterisks. Essential genes have mild but significantly lower levels of deviating noise. This is also true in the case of standard noise¹³.

E - What does the size of circles refer to, # target genes? Is that what the blob next to it is supposed to say? The caption should be more explanatory.

We clarified that the dot size reflects the number of known genes that are regulated by each Transcription Factors.

F - Wrongly referenced in caption by 'E'. Consider different colours avoiding red/green (also in Fig. S2). Some of the regression lines are solid instead of dashed - any meaning?

We changed the colors and corrected the reference. All regression lines are meant to be solid lines.

Figure S1B Caption:

Consider referring to 'standard/deviating' rather than 'absolute/relative' noise for consistency.

Done.

Figure 2:

A-E - I find the ordering of plots confusing. If it's meant as some sort of matrix view then x- and y-axis should be given by rows and columns, but this is not the case. This makes it harder to figure out, which plots to compare. Also the scaling of comparable axes differs, which makes it hard to compare trends between LB and M9.

The reviewer is correct and indeed growth of colonies in LB medium is on average twice as fast than in M9 glucose minimal agar medium. Even more prominent is the average difference in lag time. This is evident from panels A, D and E, where the scale is the same. However, if we were to use the same

scaling for M9 and LB in panels B and C, we would compromise the clarity of plots in panel C which would become much more skewed. Hence, we prefer to keep different scaling.

For example, could it be that there are two different growth rate regimes where correlation with the other variables is opposed. Case in point, correlations in Fig. 3D-F, where growth rate has been reduced, are more similar to the lowly-sampled regions in Fig. 2. These opposing trends also explain the poor correlation of growth with other variables, which is later investigated as significant in the Pareto analysis.

The point raised by the reviewer is an interesting point which we also thought about. Thanks to the analysis illustrated in figure 3, we could explore these relationship on a much wider range of growth rates, lag times and respiratory activities (including similar range as those obtained in M9 glucose minimal medium). In all conditions, we found the same type of relationships.

Consider increasing some fonts, e.g. Sp & P.

OK

Why are there different units for respiratory activity in the y-labels?

Thanks for spotting the mistake

H - When referred to from main text, what does K refer to?

We corrected the typo and refer to panel F.

Figure 4:

A - In caption, what does 'black dots' refer to?

Thanks for spotting this mistake. We removed it.

D - In caption, 'mutants' mis-spelled.

Corrected

Figure 5:

C - The colour map doesn't cover all colours shown, e.g. what do purple/blue dots mean? Same x-/y-scaling as in A would help conveying the argument that after starvation we see fewer change compared to wt.

We revised the figure and clarified that dots with a blue LineMarker highlights significant changes (qvalue<0.05). We prefer to keep the different scaling because it improves clarity of the volcano plot. However, we make explicit in the text that difference in response to starvation are much less pronounced.

D - Again, preserving scaling across plots would help comparison.

These intensities are derived from nontargeted MS analysis. Hence, intensities between different metabolites are not directly comparable.

Figure S3:

C - red line is invisible.

OK

E&F - Re-ordering the legend and considering similar colour for +/-TCC would make the case more obvious.

OK

Figure S4:

A-C - What are the numbers in the legend? CVs? Caption could be more explanatory.

OK

Figure S5:

The caption reads a bit confusing, e.g. 'scatter of individual conditions (red dots) overlaid on overall density' could be clearer.

OK

Figure S7:

What happened to the x-label?

We clarify that it represents the ratio between the two channels used for FACS measurements.

Typos:

- Introduction: "Surprisingly, we found that _that_ individual colonies..."
- After reference to Fig. 2A-D: "Coefficient of Variance" → "Coefficient of Variation"

Done

Other:

- Personally, I feel that 'SS' as abbreviation for slow switchers seems unfortunate historically and could be avoided.

To avoid possible historical misinterpretation we now changed FS and SS in FSw and SSw

Biobliography

1. Cole, J. A., Kohler, L., Hedhli, J. & Luthey-Schulten, Z. Spatially-resolved metabolic cooperativity within dense bacterial colonies. *BMC Syst. Biol.* **9**, 15 (2015).
2. Warren, M. R. *et al.* Spatiotemporal establishment of dense bacterial colonies growing on hard agar. *eLife* **8**, e41093 (2019).
3. van Vliet, S. *et al.* Spatially Correlated Gene Expression in Bacterial Groups: The Role of Lineage History, Spatial Gradients, and Cell-Cell Interactions. *Cell Syst.* **6**, 496-507.e6 (2018).
4. Phillips, N. E., Mandic, A., Omid, S., Naef, F. & Suter, D. M. Memory and relatedness of transcriptional activity in mammalian cell lineages. *Nat. Commun.* **10**, 1208 (2019).

5. Kochanowski, K. *et al.* Functioning of a metabolic flux sensor in *Escherichia coli*. *Proc. Natl. Acad. Sci.* 201202582 (2012) doi:10.1073/pnas.1202582110.
6. De Palo, G. *et al.* Adaptation as a genome-wide autoregulatory principle in the stress response of yeast. *IEE Syst. Biol.* **5**, 269–279 (2011).
7. Andersen, K. B. & Meyenburg, K. von. Are growth rates of *Escherichia coli* in batch cultures limited by respiration? *J. Bacteriol.* **144**, 114–123 (1980).
8. Basan, M. *et al.* Overflow metabolism in *Escherichia coli* results from efficient proteome allocation. *Nature* **528**, 99–104 (2015).

Reviewers' Comments:

Reviewer #1:

Remarks to the Author:

In my opinion, all reviewer comments and concerns have been addressed within the revised version of the manuscript. Some spelling and format mistakes are still available (e.g. Italic format of *Escherichia coli* in the title etc.), but I am confident that these mistakes will be eliminated during editing and proof-reading of the final paper. Thus, I recommend publication of the manuscript in its current form.

Reviewer #2:

Remarks to the Author:

I thank the authors for addressing all of my points. This is now a much stronger manuscript, and I have no further points.

Reviewer #3:

Remarks to the Author:

My main concerns have been addressed - I support publication.

- A. Weisse